# UAV and IoT-Based Systems for the Monitoring of Industrial Facilities Using Digital Twins: Methodology, Reliability Models, and Application

**DOI:** 10.3390/s22176444

**Published:** 2022-08-26

**Authors:** Yun Sun, Herman Fesenko, Vyacheslav Kharchenko, Luo Zhong, Ihor Kliushnikov, Oleg Illiashenko, Olga Morozova, Anatoliy Sachenko

**Affiliations:** 1School of Computer Science and Artificial Intelligence, Wuhan University of Technology, Wuhan 430070, China; 2School of Computer Science, Hubei University of Technology, Wuhan 430068, China; 3Department of Computer Systems, Networks and Cybersecurity, National Aerospace University “KhAI”, 17, Chkalov Str., 61070 Kharkiv, Ukraine; 4Research Institute for Intelligent Computer Systems, West Ukrainian National University, 11, Lvivska Str., 46009 Ternopil, Ukraine; 5Department of Informatics and Teleinformatics, Kazimierz Pulaski University of Technology and Humanities in Radom, ul. Malczewskiego 29, 26-600 Radom, Poland

**Keywords:** monitoring system, digital twin, unmanned aerial vehicle, reliability, multi-state system, control room, emergency control center, flying control center, private cloud group

## Abstract

This paper suggests a methodology (conception and principles) for building two-mode monitoring systems (SMs) for industrial facilities and their adjacent territories based on the application of unmanned aerial vehicle (UAV), Internet of Things (IoT), and digital twin (DT) technologies, and a set of SM reliability models considering the parameters of the channels and components. The concept of building a reliable and resilient SM is proposed. For this purpose, the von Neumann paradigm for the synthesis of reliable systems from unreliable components is developed. For complex SMs of industrial facilities, the concept covers the application of various types of redundancy (structural, version, time, and space) for basic components—sensors, means of communication, processing, and presentation—in the form of DTs for decision support systems. The research results include: the methodology for the building and general structures of UAV-, IoT-, and DT-based SMs in industrial facilities as multi-level systems; reliability models for SMs considering the applied technologies and operation modes (normal and emergency); and industrial cases of SMs for manufacture and nuclear power plants. The results obtained are the basis for further development of the theory and for practical applications of SMs in industrial facilities within the framework of the implementation and improvement of Industry 4.0 principles.

## 1. Introduction and Related Works

### 1.1. Motivation

The serviceability of control systems and monitoring of the condition of industrial facilities have emerged as a separate class of complex and ultra-complex systems. This is because of their growing importance in the context of, above all, safety; the efficiency of information; their processing and decision making to minimize the risk of costly equipment failures; the occurrence of emergency or pre-emergency situations and their prevention; and the reduction of the consequences of accidents for enterprises, the environment, as well as the residents of territories.

The relevance of improving the monitoring systems of industrial facilities has grown in recent decades for several reasons:There are a large number of emergency facilities, enterprises, and transport and energy systems that are part of regional, national, and even transnational infrastructures. Thus, their influence grows not only on the continuous and high-quality provision of relevant products, resources, and services, but also on the potential danger of certain territories and regions (more and more dangerous objects) [1,2].These systems are becoming more robotic and lacking in human participation, considering the trends in the development and implementation of the principles of Industry 4.0 [3,4]. As a result, monitoring systems need to develop appropriately, based on digital, mobile, and smart technologies; this is necessary to effectively perform control and monitoring functions on equipment and the surrounding areas; or environmental monitoring tasks, including possible pollution, forest resources, and fires (more and more robotic unmanned industries); and develop facilities for environmental monitoring (forest fires, etc.) [5,6].New technologies (Internet of Things (IoT) and Internet of Everything, big data analytics, digital twins (DTs) and X-reality, and unmanned aerial vehicles (UAVs), commonly known as drones, robotics, etc.) create new opportunities for the creation of monitoring systems; however, they are the cause of challenges related to dependability, functional safety, and cybersecurity components and monitoring systems, which operate in very harsh physical and informational conditions [7,8,9].

Therefore, it is quite important to find a solution for building monitoring systems for industrial facilities in terms of a systems approach, when these facilities, and hence, monitoring systems, are considered in certain interactions with different components (equipment, enterprises, and surrounding areas) and conditions of use.

It is also important to consider and improve the principles of monitoring industrial facilities in terms of the development and use of modern technologies, namely: digitization technologies in the context of implementing the principles of Industry 4.0, mobile and cloud technologies, X-reality technologies, security technologies, and more. A separate factor is the introduction of DT technologies [10], which are used to quickly and conveniently provide information about the status of the monitored object. In addition, the task of ensuring the dependability of monitoring systems, considering the reliability of their channels, software and hardware components, environmental impact, etc. is extremely important. This is due to the importance of the timely and trustworthy provision of information on the state of industrial facilities and their components.

### 1.2. State of the Art

To analyze the state of monitoring systems for industrial facilities research, let us *first* consider the monitoring systems based on IoT technologies and their development in the context of Industry 4.0. The Industry 4.0 paradigm encompasses many digital technologies that affect manufacturing enterprises [11]. The term “Industry 4.0” includes several key technologies such as cyber–physical systems, the IoT, artificial intelligence, big data analytics, and DTs, which can be considered the main factors of automated and digital production environments [12].

This direction is developing rapidly, allowing manufacturing companies to take advantage of new opportunities for digital transformation, and the offer products and services, including cloud-based solutions [13], in current and emerging markets at a competitive price [14]. However, there is still a lack of comprehensive research on the application of technologies that allow the utilization of Industry 4.0 in the production life cycle.

Much of the research on this topic focuses on the study of smart enterprise technologies, production planning, and relevant technological processes, and does not consider the possibility of monitoring these systems. Industry 4.0 can be considered as a broad area that includes data management, the competitiveness of production, production processes, and efficiency [13,15].

Technological developments in terms of Industry 4.0 are evolving rapidly, allowing manufacturing companies with new opportunities for digital transformation to offer products and services in current and emerging markets. However, there is a question about the impact of Industry 4.0 on the development of the industry as a whole. In [16], a sustainability-based model to assess the impact of Industry 4.0 technologies on several key performance indicators related to the sustainable development of different areas is proposed. Additionally, in [17], a study of the integration of Industry 4.0 technologies into the practice of full production service in many large manufacturing enterprises is shown.

*Secondly*, let us consider the introduction of DTs to monitoring systems in industrial facilities and the relevant issues of information integration from different sources and channels of monitoring systems. DT technology is playing an increasingly important role, and is an abstract-level concept between physical devices and external systems with the function of full simulation of the physical device’s workflow and saving the statistics of the device [18,19]. DTs are actively used to solve the problems of building systems for modeling the behavior of real objects [20,21,22]. However, this does not reduce the range of tasks that DTs solve.

The DT as a service (DTaaS) paradigm for digital transformation is proposed in [23], including intelligent scheduled maintenance, real-time monitoring, remote control, and forecasting functionality. Additionally, in [24], research was conducted on the use of DT technology in the context of the services and service systems of industrial products, and the study outlines possible applications for the stages of a closed cycle of the product’s life. In [25], the potential benefits of using the DT for the industry in terms of performance and process quality standards are shown. In addition, [26,27] explores the concept of the DT as a tool for smart services in the context of production certification, as well as real-time monitoring and forecasting. Improvements in machine learning, the IoT, and big data have made a huge contribution to improving the DT in terms of its real-time monitoring and forecasting properties.

Therefore, DT technology is currently becoming the most popular, and is used in combination with other information and communication technologies, such as augmented reality [28,29], artificial intelligence [30,31], and ontological modeling [32,33]. Authors [34,35] have shown that DTs are essential components of the SM in cases where real-time measurements are not feasible due to economical or technological factors. In these cases, DTs can be utilized for predicting such measurements during a change in the conditions of operation/facility and for allowing an operator to check whether all the predictions are in the safe zone. DTs may also be in demand for checking whether the normal operation variables are maintained in the correct range when performing optimization techniques aimed at calculating the optimal economic management of facilities [36,37,38]. Thus, DTs are widely used; however, it is best to employ them for the easy integration of data between a physical and virtual machine in any field.

Currently, the implementation of DT technologies for monitoring systems is underway [39,40,41], which are considered to have three levels: equipment, the building as a whole (several floors with equipment), and the entire adjacent area to the building with equipment. A significant number of tasks are related to the monitoring of logistics and industrial systems [42,43,44].

Along with this, the growing popularity of IoT technology has complemented the use of DT technology [45,46]. As the IoT becomes increasingly popular in intelligent environments, the concept of the DT is evolving as a complement to its physical part. Using IoT sensors, the DT collects information from a real counterpart, and then, simulates a physical object in real time, providing insights into performance and possible problems [47,48]. The digital representation of physical objects is a key component of industrial applications, as they are the basis for decision making. Thus, the conceptual approach to DTs is well-defined software that follows the entire life cycle along with their physical counterparts, from development and operation to unloading; therefore, this approach obtains a description of the type, identity, and life cycle of the object. However, considerable attention must be paid to the security of such systems [49].

*Thirdly*, let us consider using UAVs and ensuring the dependability of monitoring systems. The results of an analysis of tasks, areas of application of UAVs, and publications related to their use in monitoring systems are described in [50]. Paper [51] describes the use of Internet of Drones (IoD) technologies in UAV fleets. Paper [52] presents the results of using heterogeneous UAV fleets for monitoring in the field of nuclear energy and analyzes the advantages of heterogeneous fleets over homogeneous ones.

In [53,54], the issues of creating UAV-based monitoring systems are considered. However, it discusses the aspects of carrier selection and payload list only, and the general composition of the fleet (number, organization of management, interaction, etc.) is not determined.

Papers [55,56,57,58,59] study the management processes of monitoring systems, which are based on UAVs in various aspects, namely: self-organization in wireless networks [55], flight safety [55,56], the control and coordination of various types of robot [54], and issues of human interaction with UAVs [58,59]. Features of UAV and ground robot planning in monitoring systems and other applications are studied in [60,61,62]. However, these papers pay more attention to the aspect of flight safety, and the construction and maintenance of a given group in the fleet. Planning in UAVs, considering the indicators of their reliability and effectiveness in missions, is presented in [63,64,65]. Security issues regarding the utilization of UAV-based networks in IoT scenarios are considered in [66,67].

The group application of UAVs requires special approaches to the construction of monitoring systems as well as to their management and organization. High efficiency in the creation of such systems is provided by the introduction of multi-agent systems with features [68,69,70].

Although the design of systems with UAVs is described in many works above, their results are separate fragments. So, it cannot be used entirely for developing intelligent systems for monitoring potentially dangerous objects. This makes it necessary to conduct research related to:The development of monitoring system structures considering the different components of industrial facilities and the environment, and the application of technologies such as IoT, DTs, UAVs, etc. [5,71,72].Improving the reliability models of monitoring systems and researching their dependence on the reliability of subsystems, particularly those based on UAV fleets [73,74]; means of measuring, transmitting, and processing the information; as well as their integration using the DT.Monitoring system dependability studies, considering the possible degradation of systems due to channel failures and the corresponding reduction in the “monitoring coverage” of industrial facilities and their surrounding areas. In this case, it is advisable to use models of multi-state systems (MSSs) [75].

### 1.3. Objectives

Summing up the analysis of related works above, it should be noted that there are challenges of a theoretical and applied nature in creating a single methodology for developing architectures for monitoring systems for complex industrial facilities using UAVs, as well as models and methods for assessing the reliability of such monitoring systems. Hence, the goal of this paper is to develop the methodology for designing (building) two-mode monitoring systems for industrial objects and their adjacent territories based on the application of modern mobile and digital technologies, as well as a set of proper reliability models.

The objectives of the paper are as follows:To develop a methodology for building and creating general models (structures) of UAV-, IoT-, and DT-based monitoring systems in complex industrial facilities as multi-level and multi-state systems;To develop and explore reliability models of the monitoring systems considering applied mobile and digital technologies, operation modes (normal and emergency), etc.; To propose and discuss industrial cases of monitoring systems for manufacturing and nuclear power plants.

### 1.4. Approach and Structure

The approach to research includes the following provisions.

The conception and principles of the building and general structure of systems for monitoring industrial facilities are proposed; they are:Parts of critical infrastructures such as nuclear power plant utilities, dangerous manufacturers, oil and gas transport communications, etc.Described using a multilevel hierarchy scheme, and based on the application of the technologies:
(a)Digital twins as models of controlled sub-objects;(b)UAV fleet as an additional channel for collecting information;(c)A private cloud system as a redundant emergency center for decision-making support.


The monitoring system is analyzed as a complex system in terms of dependability considering:Various monitoring system structures, options for sub-object monitoring, and the configuration of different stationary and mobile centers for collecting information and decision making;Modes of monitoring system operation (normal and emergency modes) that vary in terms of the environment and failure rates of the components;The placement and reliability of digital twins generated by different centers and their influence on decision making;The placement and reliability of decision-making units;The processes of monitoring system degradation caused by components and channels failures. In this case, the monitoring system is addressed as an MSS.

Thus, the overall contribution of this research covers the development of the concept and principles for dependable SM building by using mobile technologies (UAVs and UAV fleets), the Industrial IoT, edge computing, and digital twins. The suggested principles allow us to implement options for the structural organization of SMs for complex industrial facilities, described using a three-tier hierarchy (equipment-utility-zone/adjacent area).

The rest of the paper is structured in the following way. The proposed concepts and principles are described in Section 2. In Section 3, the SM is analyzed and developed as a dependable system. Section 4 contains the two industrial cases: pre- and post-accident nuclear power plant monitoring systems, and a subsystem of monitoring equipment using IoT and private cloud. In Section 5, the results of the research are discussed. Section 6 highlights the conclusion and directions for future work.

## 2. Methodology for Building SMs

### 2.1. Concept of SM Building

For critical industrial facilities and monitoring systems, as their key component, there is a contradiction in the context of the development of mobile, information, and smart technologies. On the one hand, there are strict requirements for the reliability, safety, and survival of monitoring systems in industrial facilities in the pre- and post-accident period regarding the failure of sensors, communications, processing equipment, and control points. At the same time, the capabilities of UAVs and the Internet of Drones for measuring, transmitting, and processing information are growing. On the other hand, concepts and methods for creating and using reliable and resilient monitoring systems in industrial facilities under conditions of failure and accidents have not been sufficiently developed. 

To resolve this contradiction, the concept of building reliable and resilient monitoring systems is proposed, based on the development of the von Neumann paradigm of the synthesis of reliable systems from unreliable components, further based on the “system components-types of redundancy“ matrix [71,72,73]. For complex monitoring systems in industrial facilities, the concept covers the application of various types of redundancy (structural, version, time, and space) for basic components: sensors and means of converting measurement results, and means of communication, information transmission, and information processing. A UAV fleet is a key component of a redundant, multi-version, and dynamically reconfigurable SM because the application of UAVs allows for the implementation of specific kinds of structural, informational, and version redundancy that can be replenished. This increases the survivability of the SM in extreme conditions. Moreover, an additional kind of information and version redundancy is the presentation of the state of the monitored object in the form of digital twins for decision support centers. 

The concept is also complemented by the fact that to ensure the resilience of monitoring systems, the use of diversity and a spatial distribution of decision-making centers with clearly defined functions and priorities are proposed. In addition, the diversity of control methods is due to the use of DTs and cloud data processing, as well as data transfer methods using floating UAVs that form dynamically reconfigurable structures that are resistant to physical and cyber intrusion and can be recovered. The presence of mobile and protected cloud subsystems offers the possibility of dynamic and proactive reconfiguration of assets in the event of failures, physical impacts, and cyberattacks.

### 2.2. Principles of SM Building

#### 2.2.1. Three-Level Model of Object Monitoring

During the development of the structure and models of the monitoring system, a three-tier model of the presentation of the industrial facility was adopted, which includes:Equipment with sensors and critical-process management devices (equipment that is monitored and controlled, EC). Various systems of such monitoring are used in industrial facilities. They are based on wireless and IoT technologies and do not interfere with technological processes, without creating additional risks from the point of view of ensuring the safety of objects.Several systems and building complexes, within which equipment with sensors and devices for the management of critical processes (the utility that is monitored and controlled, UC) were placed. For example, for a nuclear power plant, this level covers systems and equipment whose condition is monitored by the Post-Accident Monitoring System [76].The territory of the industrial facility, which is limited by the outer perimeter, where monitoring stations (MSs) are located (zones that are monitored, ZC). For a nuclear power plant, this corresponds to the area whose condition is monitored by the Automated Radiation Monitoring System [77].

Next, we consider the option of building an industrial facility, on the territory of which there are several UCis (*i* = 1…*n*) and MSs of the same type.

#### 2.2.2. Applied Technologies

In the process of building and operating the monitoring system, the following technologies are expected to be used:DTs, which ensure the creation of digital clones in an industrial facility. This makes it possible to model and determine its state from various data on the state of its component systems and objects. The creation and use of DTs of different complexity for different crisis centers are envisaged. The model uses a digital twin instance which describes a specific object with which the twin remains associated through the life of the object. Duplicates of this type usually contain an annotated 3D model, which takes into account the measurement results received from the sensors, as well as the current and predicted values of the monitoring parameters.IoT and Internet of Flying Things to reserve wired data transmission channels from sensors and critical-process control devices, commonly equipped with previous-generation Post-Accident Monitoring Systems and Automated Radiation Monitoring Systems. Post-Accident Monitoring Systems were created in the first decade after the Fukushima accident [76,77].Edge computing technologies for data pre-processing at every level of the industrial facility monitoring model using edge nodes (EN), which make it possible to reduce data volumes, their transmission speed, and the requirements for the performance of transmission equipment. Data pre-processing also increases the efficiency of decision-making processing in crisis centers. The model of the monitoring system aims to perform boundary calculations and the placement of relevant edge nodes on both UCs and onboard UAVs (flying edge nodes, FENs).

#### 2.2.3. Redundant Centers of Control and Monitoring

To ensure the dependability and resilience of monitoring the condition of the object, crisis centers using the UAV fleet (IoD) and cloud services are utilized. The backup model provides additional centers to the usual (for example, for a nuclear power plant) control room (CR) and emergency control room (ECR):The private cloud crisis group (PCG), which processes monitoring data using cloud services and the Internet of Things;The UAV Fleet and flying control center (FCC), which receive and processes data on the condition of an industrial facility using equipment that can be placed either on one powerful UAV or under a group of UAVs (UAV fleet or IoD), can be distributed. The IoD is a kind of Internet of Flying Things based on a set of interacting drones. For the considered SMs, the IoD is a part of the flying control center (FCC) infrastructure with access to both the Private Cloud and Internet resources. Drone on-board systems collaborate with each other, ground sensors (located at the MSs), and the FCC.

Monitoring systems operate in normal and emergency modes according to the condition of the industrial facility. Additional control and monitoring centers of industrial facilities, namely PCG and FCC, are involved in the emergency mode. Under certain circumstances, they can perform certain monitoring functions in normal mode, but within the framework of this study, they only work in emergency mode.

In this case, the organization and functions of the SM based on CR, ECR, PCG, and FCC are described in Table 1 and Table 2. CR, ECR, PCG, and FCC are separate subsystems of monitoring consisting of sensors (Sen), communication (Com), data processing (Prc), and decision-making support means (DMU). Table 1 describes the functions of these subsystems considering the levels of industrial facility hierarchy (EC, UC, and ZC). Table 2 presents the capacity of monitoring functions in the normal and emergency modes.

The CR subsystem in both modes performs all monitoring functions at all levels of the hierarchy. The ECR subsystem differs in that it does not perform equipment monitoring (EC). The FCC and PCG subsystems provide monitoring at the ZC level only in emergency mode, in full or in part, depending on the sensor coverage of the surrounding areas. The ZC sensors for CR, ECR, PCG, and FCC can be general or separate.

#### 2.2.4. General Model of the Industrial Facility Monitoring System

Thus, the general model of the industrial facility monitoring system can be described as a set GSM = {OM, CM, Sen, Com, DT, DMU}, where:

OM = {EC, UC, ZC}—the set of objects that are monitored or/and controlled;

CM = {CR, ECR, PCG, FCC]—the set of centers of control and monitoring;

Sen = {Sen_EC, Sen_UC, Sen_ZC}—the set of sensors;

Com = {ComEC_CR, ComUC_CR, ComUC_ECR, ComZC_CR, ComZC_ECR, ComZC_PCG, ComZC_FCC}—the set of means of communication;

DT = {Dtw_CR, Dtw_ECR, Dtw_PCG, Dtw_FCC]—the set of digital twins;

DMU = {DMU_CR, DMU_ECR]—the set of decision-making support means.

This GSM model describes the structural components of the SM, the relationships between which are shown in Figure 1. Reliability models of the SMs are described and explored in the next sections.

### 2.3. Structures of SMs

The structures of industrial facility monitoring systems with different concepts of the use of digital twins are shown in Figure 1 and Figure 2 (types T1 and T2). The first structure (Figure 1) describes the monitoring system using DTs designated as Dtw in each crisis center: Dtw_CR in the CR center, respectively; Dtw_ECR; Dtw_PCG; and Dtw_FCC. The second structure (Figure 2) of the monitoring system involves the use of DTs only in CR and ECR—Dtw_CR and Dtw_ECR, respectively.

In the proposed structures, CR receives data on the monitoring of the condition of all the components of the three-level model of an industrial facility—EC, UC, and ZC. The transmission of monitoring information from equipment EC_ij_, i = {1…, n}, j = {1,…, m_i_} and stations MS1,…, MSk is performed either by nodes ENi and FEN, respectively, which perform data pre-processing, or directly from several UCi sensors.

ECR receives monitoring data from US1,..., UCn, and ZC. PCG and FCC only process ZC monitoring data. All the shelters have appropriate Prc tools for data processing. The decision on the condition of the facility is made only in CR and ECR; for this purpose, decision-making units (DMU) are provided in their composition: DMU_CR and DMU_ECR. All the centers interact with each other (interaction channels are indicated by a dotted line).

**Figure 2 sensors-22-06444-f002:**
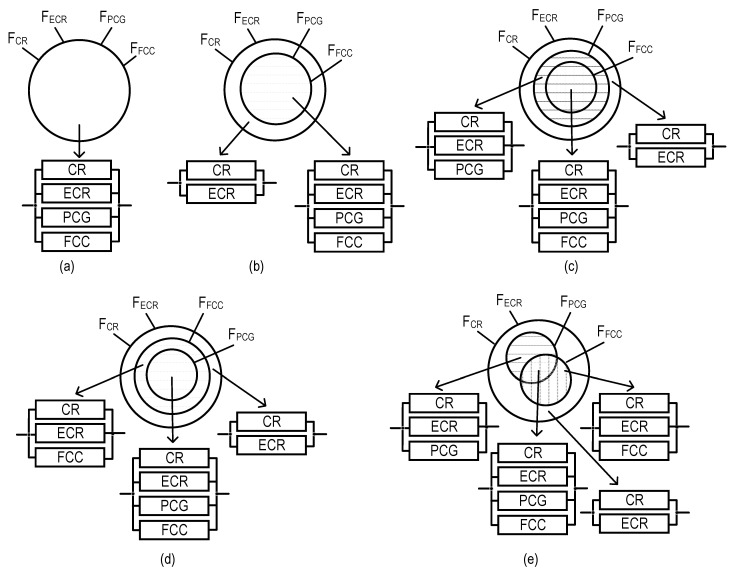
Graphical representation of models for SM option field-monitoring coverage of variants for allocation of monitoring functions among the SM channels (VAMF), as well as RBDs per each of these variants: (**a**) model for VAMF1, (**b**) model for VAMF2, (**c**) model for VAMF3, (**d**) model for VAMF4, and (**e**) model for VAMF5.

Table 3 provides options for the structures of facility monitoring systems in normal and emergency mode, taking into account data sources, relevant centers, and the type of monitoring system (T1 or T2).

Further details and research on SM structures will be given in Section 3 based on the analysis of various sensor coverage options of the ZC parameter space for PCG and FCC subsystems, and on the development of appropriate reliability models. The presence of two, three, or four monitoring subsystems, i.e., the involvement of PCG and FCC subsystems, refers to the completeness of ZC option field coverage, which may be limited for them (marked with an asterisk). The relevant coverage models are analyzed below.

## 3. Reliability Models for System of Monitoring

### 3.1. Models of SM Option Field-Monitoring Coverage

Let us assume that each SM subsystem channel comprises either communications, a processor, and a digital twin (for the type-T1 SM structure) or communications and a processor (for the type-T2 SM structure). DTs’ compatibility with different subsystems is ensured as follows:For the EC, the DTs are formed only in CR, so there is no compatibility problem here.For UC, DTs are formed in CR and ECR from common sensors, so the data for generating DTs are identical, and, therefore, their compatibility is ensured. In addition, these data are corrected by the DMU in cases of failure.For ZC, DTs are formed by all the subsystems (CR, ECR, PCG, and FCC) from sensors located in the ZC zone. If all subsystems use the full amount of monitoring data, both the compatibility and reliability of DT formation are ensured as in the previous case, using the DMU. If PCG and FCC subsystems use a limited data set from sensors, these data are employed to form the corresponding part of DTs and combine it with data from CR and ECR to ensure the DT’s compatibility.

In emergency mode, the SM can utilize either one (PCG/FCC) or two (PCG and FCC) channels in addition to the CR and ECR channels.

Depending on the tasks performed by the SM, the following features of SM channels’ functioning should be considered:Channels can perform the same set of monitoring functions;The set of monitoring functions for some channels can be considered as a subset of monitoring functions for other channels;Sets of monitoring functions for individual channels may overlap.

These features of the SM channels mentioned above can form various variants of allocation for monitoring functions (VAMFs) among the SM channels. Each VAMF allows us to obtain the corresponding model of SM option field-monitoring coverage, as well as forming reliability block diagrams (RBDs). The latter are needed to assess the reliability of SM, which uses such VAMFs.

Let us introduce the following notations: F_CR_, F_ECR_, F_PCG_, and F_FCC_ are sets of monitoring functions performed by the CR, ECR, PCG, and FCC channels, respectively.

Considering the presented features of SM channel functioning and the accepted notations, five models of SM option field-monitoring coverage were developed (Figure 2).

As we can see from Figure 2, the largest and lowest numbers of RBDs correspond to VAMF5 (four RBDs, Figure 2e) and VAMF1 (one RBD, Figure 2a), respectively.

### 3.2. Extended Specification for SM Structures

Considering the specification of the SM structures presented in Table 3 and the features of the models of SM option field-monitoring coverage for VAMF, an extended specification was developed (Table 4).

This specification, in addition to the source of information (EC, UC, or ZC), mode of functioning (N or E), and type of structure (T1 or T2), which are presented in Table 3, provides information on:−The third channel for the three-channel SM structure (PCG or FCC);−The designation of the SM structure;−The reliability block diagram for the SM;−The notations of the SM structure and SM reliability function;−The equation for calculating the SM reliability function.

**Table 4 sensors-22-06444-t004:** Extended specification of the SM structures.

Source of Information	Mode	Number of Channels	Type of Structure	Third Channel for Three-Channel SM structure	Notation of SM Structure	Reliability Block Diagram for SM	Notation of SM Reliability Function	Equation for Calculating SM Reliability Function
EC	Normal	1	T2	-	$S_{1}^{EC\_N}$	Figure 3	$P_{1}^{EC\_N}$	Equation (1)
EC	Emergency	1	T2	-	$S_{1}^{EC\_E}$	Figure 3	$P_{1}^{EC\_E}$	Equation (2)
UC	Normal	2	T1	-	$S_{2\_1}^{UC\_N}$	Figure 4	$P_{2\_1}^{UC\_N}$	Equation (3)
UC	Emergency	2	T1	-	$S_{2\_1}^{UC\_E}$	Figure 4	$P_{2\_1}^{UC\_E}$	Equation (4)
UC	Normal	2	T2	-	$S_{2\_2}^{UC\_N}$	Figure 5	$P_{2\_2}^{UC\_N}$	Equation (5)
UC	Emergency	2	T2	-	$S_{2\_2}^{UC\_E}$	Figure 5	$P_{2\_2}^{UC\_E}$	Equation (6)
ZC	Normal	2	T1	-	$S_{2\_1}^{ZC\_N}$	Figure 6	$P_{2\_1}^{ZC\_N}$	Equation (7)
ZC	Emergency	2	T1	-	$S_{2\_1}^{ZC\_E}$	Figure 6	$P_{2\_1}^{ZC\_E}$	Equation (8)
ZC	Normal	2	T2	-	$S_{2\_2}^{ZC\_N}$	Figure 7	$P_{2\_2}^{ZC\_N}$	Equation (9)
ZC	Emergency	2	T2	-	$S_{2\_2}^{ZC\_E}$	Figure 7	$P_{2\_2}^{ZC\_E}$	Equation (10)
ZC	Emergency	3	T1	PCG	$S_{3\left(PCG\right)\_1}^{ZC\_E}$	Figure 8	$P_{3\left(PCG\right)\_1}^{ZC\_E}$	Equation (11)
ZC	Emergency	3	T2	PCG	$S_{3\left(PCG\right)\_2}^{ZC\_E}$	Figure 9	$P_{3\left(PCG\right)\_2}^{ZC\_E}$	Equation (12)
ZC	Emergency	3	T1	FCC	$S_{3\left(FCC\right)\_1}^{ZC\_E}$	Figure 10	$P_{3\left(FCC\right)\_1}^{ZC\_E}$	Equation (13)
ZC	Emergency	3	T2	FCC	$S_{3\left(FCC\right)\_2}^{ZC\_E}$	Figure 11	$P_{3\left(FCC\right)\_2}^{ZC\_E}$	Equation (14)
ZC	Emergency	4	T1	-	$S_{4\_1}^{ZC\_E}$	Figure 12	$S_{4\_1}^{ZC\_E}$	Equation (15)
ZC	Emergency	4	T2	-	$S_{4\_2}^{ZC\_E}$	Figure 13	$P_{4\_2}^{ZC\_E}$	Equation (16)

### 3.3. Reliability Models

#### 3.3.1. Notations and Assumptions

The used notations are as follows:

Com*α*_*β* is the communications between *α* and *β*, where *α* = EC, UC, ZC and *β* = CR, ECR, PCG, FCC.

DMU_*γ* is the decision-making support means in *γ*, where *γ* = CR, ECR.

Dtw_*β*/Prc_*β* is the digital twins/data processing means in *β*, where *β* = CR, ECR, PCG, FCC.

$P_{i}$ is the reliability function of *I*, where *i* = Com*α*_*β*, DMU_*γ*, Dtw_*β*, Prc_*β*; *α* = EC, UC, ZC; *β* = CR, ECR, PCG, FCC; and *γ* = CR, ECR.

t is the operating time.

λ is the basic failure rate corresponding to the failure rate of ComFCC.

$k_{i}$ is a coefficient by which the failure rates of Com*α*_CR and Com*ψ*_ECR must be multiplied to obtain their failure rates for the emergency mode, where *α* = EC, UC, ZC and *ψ* = EC, UC, ZC.

$k_{E}$ is the coefficient by which the failure rates of ComEC_CR, ComUC_CR, ComZC_CR, ComUC_ECR, and ComZC_ECR must be multiplied to obtain their failure rates for the emergency mode.

2$k_{E}$ is the coefficient by which the failure rate of ComZC_*ω* must be multiplied to obtain its failure rate for the emergency mode, where *ω* = PCG, FCC.

The used assumptions are as follows:Components of the SM have an exponential time to failure;During the operating time, the SM is considered an unrecoverable system.

Limited computing capabilities and limited battery life are taken into account when choosing a UAV based on the monitoring tasks and operating conditions of the SM. The examples in this Section show how to ensure the reliability of UAVs and their batteries in the case of limited resources.

#### 3.3.2. Description and Simulation of the Reliability Models

Block diagrams and equations for SM reliability assessment (see Table 4) are presented below.
Figure 3RBD that is utilized to calculate the reliability functions of systems S1EC_N and S1EC_E using Equations (1) and (2), respectively.
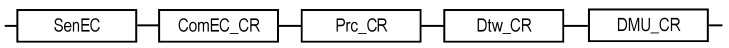

(1)$$P_{1}^{EC\_N}=P_{SenEC}P_{ComEC\_CR}P_{Prc\_CR}P_{Dtw\_CR}P_{DMU\_CR}$$
where $P_{SenEC}=e^{-\lambda k_{SenEC}t}$, $P_{ComEC\_CR}=e^{-\lambda k_{ComEC\_CR}t}$, $P_{Prc\_CR}=e^{-\lambda k_{Prc\_CR}t}$, $P_{Dtw\_CR}=e^{-\lambda k_{Dtw\_CR}t}\;,\;P_{DMU\_CR}=e^{-\lambda k_{DMU\_CR}t}$.
(2)$$P_{1}^{EC\_E}=P_{SenEC}P_{ComEC\_CR}^{E}P_{Prc\_CR}P_{Dtw\_CR}P_{DMU\_CR}$$
where $P_{ComEC\_CR}^{E}=e^{-k_{E}\lambda k_{ComEC\_CR}t}$.
Figure 4RBD that is utilized to calculate the reliability functions of systems S2_1UC_N and S2_1UC_E using Equations (3) and (4), respectively.
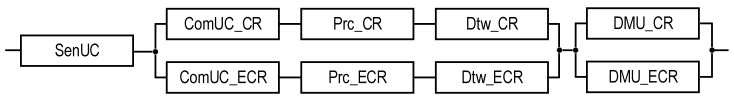

(3)$$P_{2\_1}^{UC\_N}=P_{SenUC}\left(1-\left(1-P_{ComUC_{CR}}\;P_{Prc_{CR}}P_{Dtw_{CR}}\right)\left(1-P_{ComUC\_ECR}\;P_{Prc\_ECR}P_{Dtw\_ECR}\right)\right)\;\times \left(1-\left(1-P_{DMU\_CR}\right)\;\left(1-P_{DMU\_ECR}\right)\right)$$
where $P_{SenUC}=e^{-\lambda k_{SenUC}t},$
$P_{ComUC\_CR}=e^{-\lambda k_{ComUC\_CR}t}$, $P_{ComUC\_ECR}=e^{-k_{E}\lambda k_{ComUC\_ECR}t}$, $P_{Prc\_ECR}=e^{-k_{E}\lambda k_{Prc\_ECR}t}$, $P_{Dtw\_ECR}=e^{-\lambda k_{Dtw\_ECR}t},\;P_{DMU\_ECR}=e^{-\lambda k_{DMU\_ECR}t}$.
(4)$$P_{2\_1}^{UC\_E}=P_{SenUC}^{E}\left(1-\left(1-P_{ComUC\_CR}^{E}\;P_{Prc\_CR}P_{Dtw\_CR}\right)\left(1-P_{ComUC\_ECR}^{E}\;P_{Prc\_ECR}P_{Dtw\_ECR}\right)\right)\;\times \left(1-\left(1-P_{DMU\_CR}\right)\left(1-P_{DMU\_ECR}\right)\right)$$
where $P_{ComUC\_CR}^{E}=e^{-k_{E}\lambda k_{ComUC\_CR}t}$, $P_{ComUC\_ECR}^{E}=e^{-k_{E}\lambda k_{ComUC\_ECR}t}$.
Figure 5RBD that is utilized to calculate the reliability functions of systems S2_2UC_N and S2_2UC_E using Equations (5) and (6), respectively.
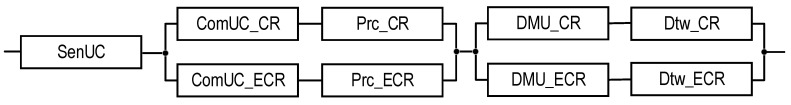

(5)$$P_{2\_2}^{UC\_N}=P_{SenUC}\left(1-\left(1-P_{ComUC\_CR}\;P_{Prc\_CR}\right)\left(1-P_{ComUC\_ECR}\;P_{Prc\_ECR}\right)\right)\left(1\;-\left(1-P_{DMU\_CR}P_{Dtw\_CR}\right)\left(1-P_{DMU\_ECR}P_{Dtw\_ECR}\right)\right)$$
(6)$$P_{2\_2}^{UC\_E}=P_{SenUC}\left(1-\left(1-P_{ComUC\_CR}^{E}\;P_{Prc\_CR}\right)\left(1-P_{ComUC\_ECR}^{E}\;P_{Prc\_ECR}\right)\right)\left(1\;-\left(1-P_{DMU\_CR}P_{Dtw\_CR}\right)\left(1-P_{DMU\_ECR}P_{Dtw\_ECR}\right)\right)$$
Figure 6RBD that is utilized to calculate the reliability functions of systems S2_1ZC_N and S2_1ZC_E using Equations (7) and (8), respectively.
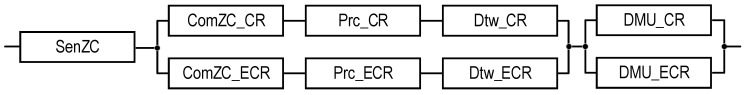

(7)$$P_{2\_1}^{ZC\_N}=P_{SenZC}\left(1-\left(1-P_{ComZC\_CR}\;P_{Prc\_CR}P_{Dtw\_CR}\;\right)\left(1-P_{ComZC\_ECR}\;P_{Prc\_ECR}P_{Dtw\_ECR}\right)\right)\;\times \left(1-\left(1-P_{DMU\_CR}\right)\;\left(1-P_{DMU\_ECR}\;\right)\;\right)$$
where $P_{SenZC}=e^{-\lambda k_{SenZC}t},$
$P_{ComZC\_CR}=e^{-\lambda k_{ComZC\_CR}t}$, $P_{ComZC\_ECR}=e^{-\lambda k_{ComZC\_ECR}t}$
(8)$$P_{2\_1}^{ZC\_E}=P_{SenZC}\left(1-\left(1-P_{ComZC\_CR}^{E}\;P_{Prc\_CR}P_{Dtw\_CR}\;\right)\left(1-P_{ComZC\_ECR}^{E}\;P_{Prc\_ECR}P_{Dtw\_ECR}\right)\right)\;\times \left(1-\left(1-P_{DMU\_CR}\right)\left(1-P_{DMU\_ECR}\right)\right)$$
where $P_{ComZC\_ECR}^{E}=e^{-k_{E}\lambda k_{ComZC\_ECR}t}$
Figure 7RBD that is utilized to calculate the reliability functions of systems S2_2ZC_N and S2_2ZC_E using Equations (9) and (10), respectively.
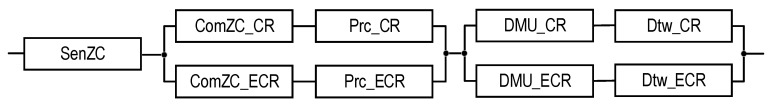

(9)$$P_{2\_2}^{ZC\_N}=P_{SenZC}\left(1-\left(1-P_{ComZC\_CR}\;P_{Prc\_CR}\right)\left(1-P_{ComZC\_ECR}\;P_{Prc\_ECR}\right)\right)\left(1\;-\left(1-P_{DMU\_CR}P_{Dtw\_CR}\right)\left(1-P_{DMU\_ECR}P_{Dtw\_ECR}\right)\;\right)$$
(10)$$P_{2\_2}^{ZC\_E}=P_{SenZC}\left(1-\left(1-P_{ComZC\_CR}^{E}\;P_{Prc\_CR}\right)\left(1-P_{ComZC\_ECR}^{E}\;P_{Prc\_ECR}\right)\right)\left(1\;-\left(1-P_{DMU\_CR}P_{Dtw\_CR}\right)\left(1-P_{DMU\_ECR}P_{Dtw\_ECR}\right)\right)$$
Figure 8RBD that is utilized to calculate the reliability function of system S3(PCG)_1ZC_E using Equation (11).
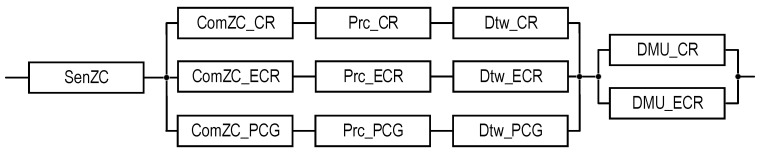

(11)$$P_{3\left(PCG\right)\_1}^{ZC\_E}=P_{SenZC}\left(1-\left(1-P_{ComZC\_CR}^{E}\;P_{Prc\_CR}P_{Dtw\_CR}\right)\left(1-P_{ComZC\_ECR}^{E}\;P_{Prc\_ECR}P_{Dtw\_ECR}\right)\times \left(1-\;P_{ComZC\_PCG}^{E}\;P_{Prc\_PCG}P_{Dtw\_PCG}\right)\right)\left(1-\left(1-P_{DMU\_CR}\right)\left(1-P_{DMU\_ECR}\right)\right)$$
where $P_{ComZC\_PCG}^{E}=e^{-2k_{E}\lambda k_{ComZC\_PCG}t},$
$P_{Prc\_PCG}=e^{-\lambda k_{Prc\_PCG}t},$
$P_{Dtw\_PCG}^{E}=e^{-\lambda k_{Dtw\_PCG}t}$.
Figure 9RBD that is utilized to calculate the reliability function of system S3(PCG)_2ZC_E using Equation (12).
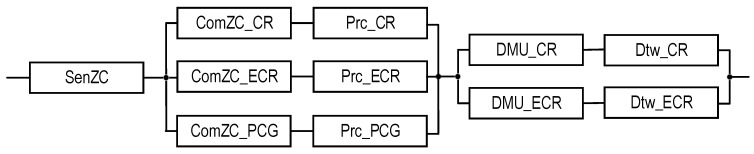

(12)$$P_{3\left(PCG\right)\_2}^{ZC\_E}=P_{SenZC}\left(1-\left(1-P_{ComZC\_CR}^{E}\;P_{Prc\_CR}\right)\left(1-P_{ComZC\_ECR}^{E}\;P_{Prc\_ECR}\right)\left(1-P_{ComZC\_PCG}^{E}\;P_{Prc\_PCG}\right)\right)\left(1\;-\left(1-P_{DMU\_CR}P_{DtwZC\_CR}\right)\left(1-P_{DMU\_ECR}P_{Dtw\_ECR}\right)\right)$$
Figure 10RBD that is utilized to calculate reliability function of system S3(FCC)_1ZC_E using Equation (13).
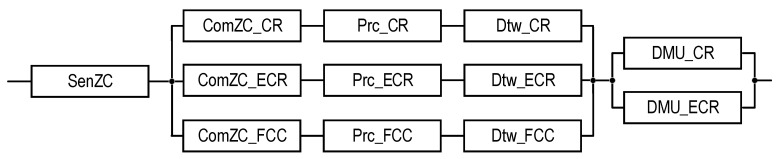

(13)$$P_{3\left(FCC\right)\_1}^{ZC\_E}=P_{SenZC}\left(1-\left(1-P_{ComZC\_CR}^{E}\;P_{Prc\_CR}P_{Dtw\_CR}\right)\;\left(1-P_{ComZC\_ECR}^{E}\;P_{Prc\_ECR}P_{Dtw\_ECR}\right)\times \left(1-\;P_{ComZC\_FCC}^{E}\;P_{Prc\_FCC}P_{Dtw\_FCC}\right)\right)\left(1-\left(1-P_{DMU\_CR}\right)\left(1-P_{DMU\_ECR}\right)\right)$$
where $P_{ComZC\_FCC}^{E}=e^{-2k_{E}\lambda k_{ComZC\_FCC}t}$, $P_{Prc\_FCC}^{E}=e^{-\lambda k_{Prc\_FCC}t}$, $P_{Dtw\_FCC}^{E}=e^{-\lambda k_{Dtw\_FCC}t}$
Figure 11RBD that is utilized to calculate reliability function of system S3(FCC)_2ZC_E using Equation (14).
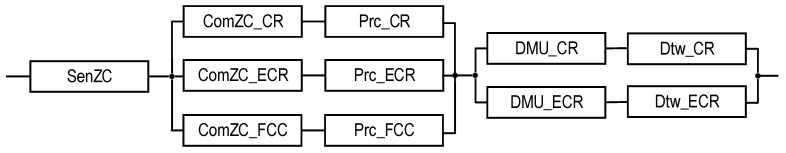

(14)$$P_{3\left(FCC\right)\_2}^{ZC\_E}=P_{SenZC}\left(1-\left(1-P_{ComZC\_CR}^{E}\;P_{Prc\_CR}\right)\left(1-P_{ComZC\_ECR}^{E}\;P_{Prc\_ECR}\right)\left(1-P_{ComZC\_FCC}^{E}\;P_{Prc\_FCC}\right)\;\right)\left(1\;-\left(1-P_{DMU\_CR}P_{DtwZC\_CR}\right)\left(1-P_{DMU\_ECR}P_{Dtw\_ECR}\right)\;\right)$$
Figure 12RBD that is utilized to calculate the reliability function of system S4_1ZC_E using Equation (15).
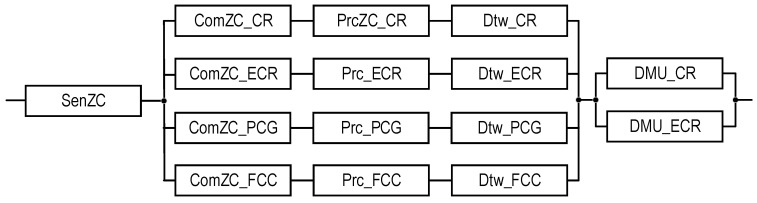

(15)$$P_{4\_1}^{ZC\_E}=P_{SenZC}\left(1-\left(1-P_{ComZC\_CR}^{E}\;P_{Prc\_CR}P_{Dtw\_CR}\right)\left(1-P_{ComZC\_ECR}^{E}\;P_{Prc\_ECR}P_{Dtw\_ECR}\right)\left(1-\;P_{ComZC\_PCG}^{E}\;P_{Prc\_PCG}P_{Dtw\_PCG}\right)\left(1-P_{ComZC\_FCC}^{E}\;P_{Prc\_FCC}P_{Dtw\_FCC}\right)\right)\left(1-\left(1-P_{DMU\_CR}\right)\left(1-P_{DMU\_ECR}\right)\;\right)$$
Figure 13RBD that is utilized to calculate the reliability function of system S4_2ZC_E using Equation (16).
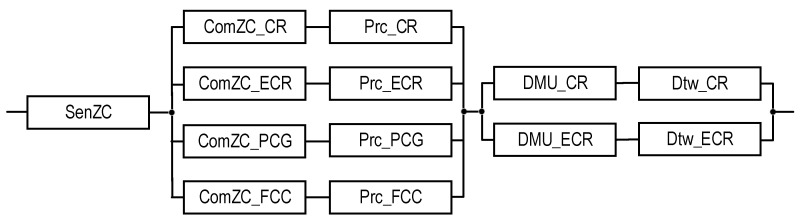

(16)$$P_{4\_2}^{ZC\_E}=P_{SenZC}\left(1-\left(1-P_{ComZC\_CR}^{E}\;P_{Prc\_CR}\right)\left(1-P_{ComZC\_ECR}^{E}\;P_{Prc\_ECR}\right)\left(1-P_{ComZC\_PCG}^{E}\;P_{Prc\_PCG}\right)\left(1-\;P_{ComZC\_FCC}^{E}\;P_{Prc\_FCC}\right)\right)\left(1-\left(1-P_{DMU\_CR}P_{DtwZC\_CR}\right)\left(1-P_{DMU\_ECR}P_{Dtw\_ECR}\right)\right)$$

Using Equations (1)–(16), some dependencies were obtained (Figure 14 and Figure 15) where the initial data are as follows:$\lambda =0.001\frac{1}{h}$, $k_{SenZC}=0.0001$, $k_{ComZC\_CR}=0.2,$
$k_{ComZC\_ECR}=0.5,$
$k_{ComZC\_FCC}=1,$
$k_{Prc\_CR}=0.01,$
$k_{Prc\_ECR}=0.01,$
$k_{Prc\_FCC}=0.1,$
$k_{Dtw\_CR}=0.001,$
$k_{Dtw\_ECR}=0.001,$
$k_{Dtw\_PCG}=0.0003,$
$k_{Dtw\_FCC}=0.0005$, $k_{DMU\_CR}=0.00001,$ and $k_{DMU\_ECR}=$ 0.00001.

**Figure 14 sensors-22-06444-f014:**
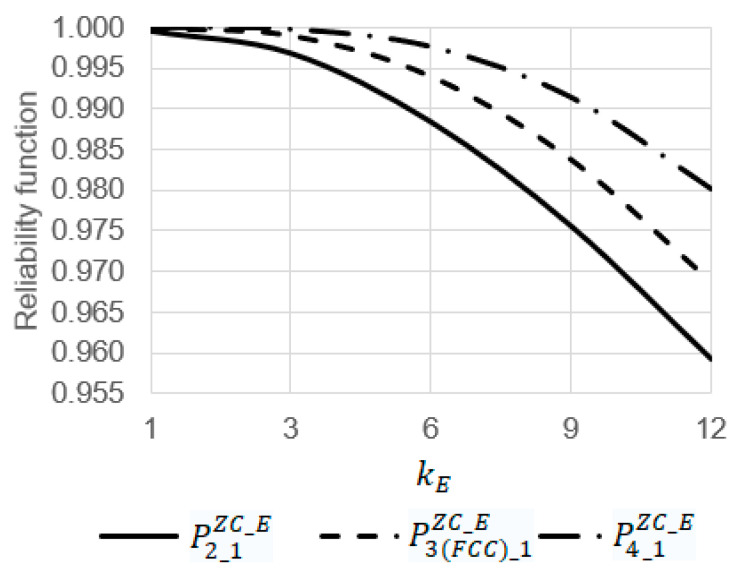
Dependency of reliability functions $P_{2\_1}^{ZC\_E}$, $P_{3\left(FCC\right)\_1}^{ZC\_E}$, and $P_{4\_1}^{ZC\_E}$ on the emergency coefficient $k_{E}$ at the operating time t = 6 h.

**Figure 15 sensors-22-06444-f015:**
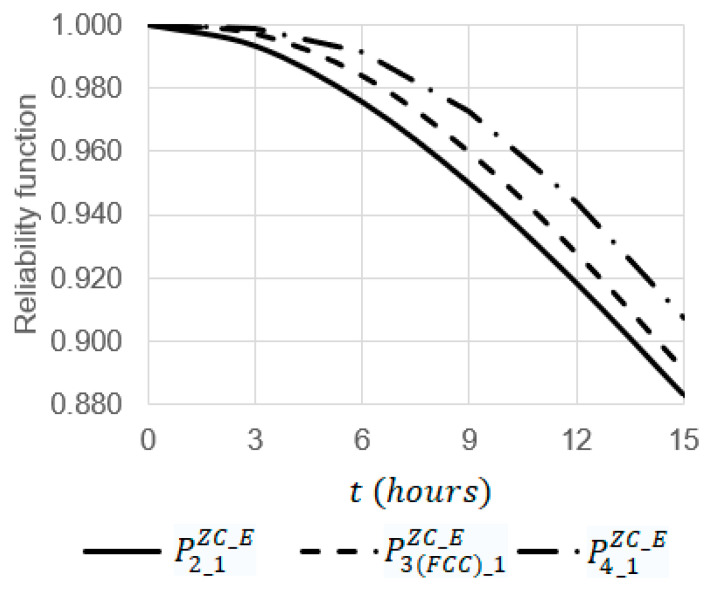
Dependency of reliability functions $P_{2\_1}^{ZC\_E}$, $P_{3\left(FCC\right)\_1}^{ZC\_E}$, and $P_{4\_1}^{ZC\_E}$ on the operating time t at the emergency coefficient $k_{E}$ = 9.

The analysis of the dependencies obtained allowed us to draw the following conclusions.
At t=6 h, the increase in the emergency coefficient $\;k_{E}$ from 1 to 12 leads to a decrease in the values of the reliability functions $P_{2\_1}^{ZC\_E}$, $P_{3\left(FCC\right)\_1}^{ZC\_E}$, and $P_{4\_1}^{ZC\_E}$ 1.04 (from 0.99961 to 0.95921), 1.03 (from 0.99996 to 0.97409), and 1.02 (from 0.99999 to 0.98019) times, respectively (Figure 14);At $k_{E}=9$, the increase in the operating time  t from 0 to 15 leads to a decrease in the values of the reliability functions $P_{2\_1}^{ZC\_E}$, $P_{3\left(FCC\right)\_1}^{ZC\_E}$, and $P_{4\_1}^{ZC\_E}$ 1.13 (from 1 to 0.88303), 1.12 (from 1 to 0.89077), and 1.1 (from 1 to 0.90725) times, respectively (Figure 15);Among the systems $S_{2\_1}^{ZC\_E}$, $S_{3\left(FCC\right)\_1}^{ZC\_E}$, and $S_{4\_1}^{ZC\_E}$, the most reliable system is $S_{4\_1}^{ZC\_E}$ and the most unreliable one is $S_{2\_1}^{ZC\_E}$. For example, at $k_{E}=9$ and t=12 h, the value of the reliability function $P_{4\_1}^{ZC\_E}$ is 1.02 times larger than the value of the reliability function $P_{3\left(FCC\right)\_1}^{ZC\_E}$ (0.94358 against 0.92764) and 1.03 times larger than the value of the reliability function $P_{2\_1}^{ZC\_E}$ (0.94358 against 0.92764) (Figure 15).

### 3.4. Models of SM as a Multi-State System

#### 3.4.1. Description of SM as a Multi-State System

In the emergency mode, an SM utilizing some VAMFs and the Sen_ZC as a source of information can be in more than one operable state; in other words, it can be considered an MSS [70]. Graphical depictions of operable states of an SM utilizing VAMF2 (system $S_{VAMF2}^{ZC\_E}$), VAMF3 ($S_{VAMF3}^{ZC\_E}$), and VAMF4 ($S_{VAMF4}^{ZC\_E}$) are shown further in Figures 23, 24, and 25, respectively.

In Figure 16, Figure 17 and Figure 18, FSM is a set of monitoring functions performed by the SM. 

The system $S_{VAMF2}^{ZC\_E}$ (Figure 16) has two operable states (the fully operable state (L1) and the partially operable state (L2)), while both $S_{VAMF3}^{ZC\_E}$ (Figure 17) and $S_{VAMF4}^{ZC\_E}$ (Figure 18) have free operable states (L1, L2, and the partially operable state, L3). Each state can be characterized by an RBD comprising binary-state channels. The operable and non-operable states of the channels are shown in white and gray, respectively.

**Figure 16 sensors-22-06444-f016:**
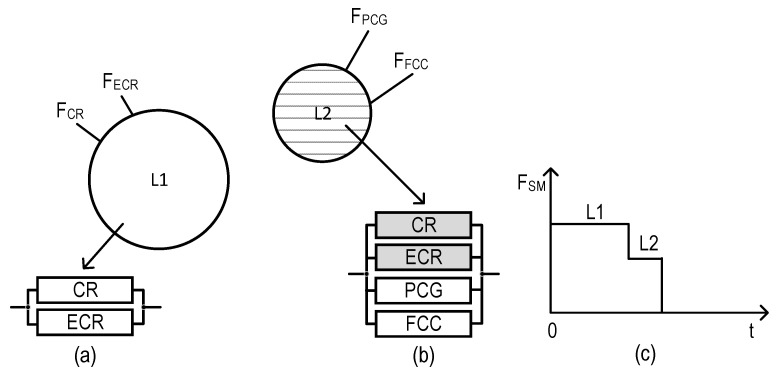
Graphical depiction of operable states for the system $S_{VAMF2}^{ZC\_E}$: (**a**) sets of monitoring functions and RBDs which correspond to state L1, (**b**) sets of monitoring functions and RBDs which correspond to state L2, and (**c**) SM degradation diagram.

**Figure 17 sensors-22-06444-f017:**
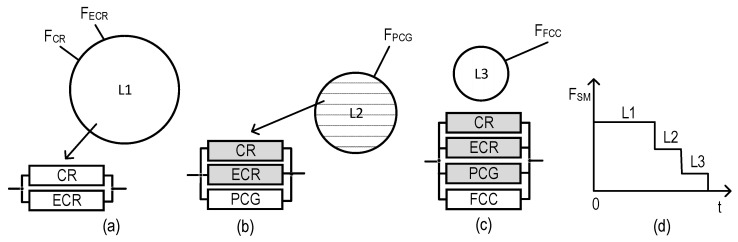
Graphical depiction of operable states for the system $S_{VAMF3}^{ZC\_E}$: (**a**) sets of monitoring functions and RBDs which correspond to state L1, (**b**) set of monitoring functions and RBDs which correspond to state L2, (**c**) set of monitoring functions and RBDs which correspond to state L3, and (**d**) SM degradation diagram.

**Figure 18 sensors-22-06444-f018:**
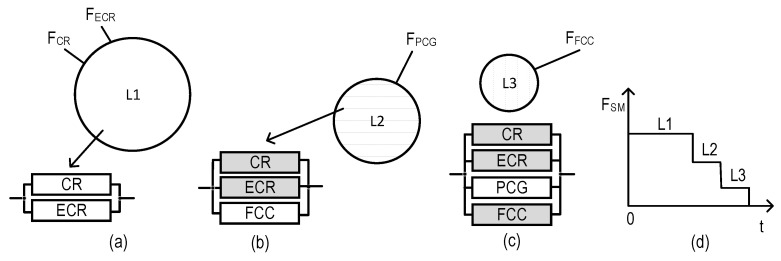
Graphical depiction of operable states for the system $S_{VAMF4}^{ZC\_E}$: (**a**) sets of monitoring functions and RBDs which correspond to state L1, (**b**) set of monitoring functions and RBDs which correspond to state L2, (**c**) set of monitoring functions and RBDs which correspond to state L3, and (**d**) SM degradation diagram.

#### 3.4.2. Reliability Models

Based on Figure 16, the probabilities of the system $S_{VAMF2}^{ZC\_E}$ being at the given states can be calculated using Equations (17)–(19).
(17)$$P_{VAMF2\_L1}^{ZC\_E}=P_{SenZC}\left(1-\left(1-P_{ZC\_CR}^{E}\right)\left(1-P_{ZC\_ECR}^{E}\right)\right)\left(1-\left(1-P_{DMU\_CR}\right)\left(1-P_{DMU\_ECR}\right)\right)$$
where $P_{VAMF2\_L1}^{ZC\_E}$ is the probability of the system $S_{VAMF2}^{ZC\_E}$ being at state L1, $P_{ZC\_CR}^{E}=P_{ComZC\_CR}^{E}\;P_{Prc\_CR}P_{Dtw\_CR}$, $P_{ZC\_ECR}^{E}=P_{ComZC\_ECR}^{E}\;P_{Prc\_ECR}P_{Dtw\_ECR}$.
(18)$$P_{VAMF2\_L2}^{ZC\_E}=P_{SenZC}\left(1-P_{ZC\_CR}^{E}\right)\left(1-P_{ZC\_ECR}^{E}\right)\left(1-\left(1-P_{ZC\_PCG}^{E}\right)\left(1-P_{ZC\_FCC}^{E}\right)\right)\left(1\;-\left(1-P_{DMU\_CR}\right)\left(1-P_{DMU\_ECR}\right)\right)$$
where $P_{VAMF2\_L2}^{ZC\_E}$ is the probability of the system $S_{VAMF2}^{ZC\_E}$ being at state L2, $P_{ZC\_PCG}^{E}=P_{ComZC\_PCG}^{E}\;P_{Prc\_PCG}P_{Dtw\_PCG}$, $P_{ZC\_FCC}^{E}=P_{ComZC\_FCC}^{E}\;P_{Prc\_FCC}P_{Dtw\_FCC}$.
(19)$$P_{VAMF2\geq L2}^{ZC\_E}=P_{VAMF2\_L1}^{ZC\_E}+P_{VAMF2\_L2}^{ZC\_E}$$
where $P_{VAMF2\_\geq L2}^{ZC\_E}$ is the probability of the system $S_{VAMF2}^{ZC\_E}$ being at state L2 or above.

Based on Figure 17, the probabilities of the system $S_{VAMF3}^{ZC\_E}$ being at the given states can be calculated using Equations (20)–(24).
(20)$$P_{VAMF3\_L1}^{ZC\_E}=P_{SenZC}\left(1-\left(1-P_{ZC\_CR}^{E}\right)\left(1-P_{ZC\_ECR}^{E}\right)\right)\left(1-\left(1-P_{DMU\_CR}\right)\left(1-P_{DMU\_ECR}\right)\right)$$
where $P_{VAMF3\_L1}^{ZC\_E}$ is the probability of the system $S_{VAMF3}^{ZC\_E}$ being at state L1.
(21)$$P_{VAMF3\_L2}^{ZC\_E}=P_{SenZC}\left(1-P_{ZC\_CR}^{E}\right)\left(1-P_{ZC\_ECR}^{E}\right)P_{ZC\_PCG}^{E}\left(1-\left(1-P_{DMU\_CR}\right)\left(1-P_{DMU\_ECR}\right)\right)$$
where $P_{VAMF3\_L2}^{ZC\_E}$ is the probability of the system $S_{VAMF3}^{ZC\_E}$ being at state L2.
(22)$$P_{VAMF3\_L3}^{ZC\_E}=P_{SenZC}\left(1-P_{ZC\_CR}^{E}\right)\left(1-P_{ZC\_ECR}^{E}\right)\left(1-P_{ZC\_PCG}^{E}\right)P_{ZC\_FCC}^{E}\left(1-\left(1-P_{DMU\_CR}\right)\left(1-P_{DMU\_ECR}\right)\right)$$
where $P_{VAMF3\_L3}^{ZC\_E}$ is the probability of the system $S_{VAMF3}^{ZC\_E}$ being at state L3.
(23)$$P_{VAMF3\_\geq L2}^{ZC\_E}=P_{VAMF3\_L1}^{ZC\_E}+P_{VAMF3\_L2}^{ZC\_E}$$
where $P_{VAMF3\_\geq L2}^{ZC\_E}$ is the probability of the system $S_{VAMF3}^{ZC\_E}$ being at state L2 or above.
(24)$$P_{VAMF3\_\geq L3}^{ZC\_E}=P_{VAMF3\_L1}^{ZC\_E}+P_{VAMF3\_L2}^{ZC\_E}+P_{VAMF3\_L3}^{ZC\_E}$$
where $P_{VAMF3\_\geq L3}^{ZC\_E}$ is the probability of the system $S_{VAMF3}^{ZC\_E}$ being at state L3 or above.

Based on Figure 18, the probabilities of the system $S_{VAMF4}^{ZC\_E}$ being at the given states can be calculated by Equations (25)–(29).
(25)$$P_{VAMF4\_L1}^{ZC\_E}=P_{SenZC}\left(1-\left(1-P_{ZC\_CR}^{E}\right)\left(1-P_{ZC\_ECR}^{E}\right)\right)\left(1-\left(1-P_{DMU\_CR}\right)\left(1-P_{DMU\_ECR}\right)\right)$$
where $P_{VAMF4\_L1}^{ZC\_E}$ is the probability of the system $S_{VAMF4}^{ZC\_E}$ being at state L1.
(26)$$P_{VAMF4\_L2}^{ZC\_E}=P_{SenZC}\left(1-P_{ZC\_CR}^{E}\right)\left(1-P_{ZC\_ECR}^{E}\right)P_{ZC\_FCC}^{E}\left(1-\left(1-P_{DMU\_CR}\right)\left(1-P_{DMU\_ECR}\right)\right)$$
where $P_{VAMF4\_L2}^{ZC\_E}$ is the probability of the system $S_{VAMF4}^{ZC\_E}$ being at state L2.
(27)$$P_{VAMF4\_L3}^{ZC\_E}=P_{SenZC}^{E}\left(1-P_{ZC\_CR}^{E}\right)\left(1-P_{ZC\_ECR}^{E}\right)\left(1-P_{ZC\_FCC}^{E}\right)P_{ZC\_PCG}^{E}\left(1-\left(1-P_{DMU\_CR}\right)\left(1-P_{DMU\_ECR}\right)\right)$$
where $P_{VAMF4\_L3}^{ZC\_E}$ is the probability of the system $S_{VAMF4}^{ZC\_E}$ being at state L3.
(28)$$P_{VAMF4\_\geq L2}^{ZC\_E}=P_{VAMF4\_L1}^{ZC\_E}+P_{VAMF4\_L2}^{ZC\_E}$$
where $P_{VAMF4\_\geq L2}^{ZC\_E}$ is the probability of the system $S_{VAMF4}^{ZC\_E}$ being at state L2 or above.
(29)$$P_{VAMF4\_\geq L3}^{ZC\_E}=P_{VAMF4\_L1}^{ZC\_E}+P_{VAMF4\_L2}^{ZC\_E}+P_{VAMF4\_L3}^{ZC\_E}$$

#### 3.4.3. Simulation and Analysis

For the simulation, the system $S_{VAF3}^{ZC\_E}$ (Figure 17) was chosen. Using Equations (17)–(29), some dependencies were obtained (Figure 19, Figure 20 and Figure 21) where the initial data were the same, as they were used for obtaining the dependencies presented in Section 3.1.

The analysis of the dependencies obtained allowed us to draw the following conclusions.

For $k_{E}=12$. (see Figure 19):
The increase in the operating *t* time from 0 to 15 leads to a decrease in the probabilities of $P_{VAMF3\_\geq L3}^{ZC\_E}$, $P_{VAMF3\_\geq L2}^{ZC\_E}$, and $P_{VAMF3\_L1}^{ZC\_E}$ by 1.19 (from 1 to 0.83865), 1.20 (from 1 to 0.83419), and 1.21 (from 1 to 0.81969), respectively;At t=15 h, the probability of $P_{VAMF3\_\geq L3}^{ZC\_E}$ is 1.01 times larger than the probability of $P_{VAMF3\_\geq L2}^{ZC\_E}$, (0.83865 against 0.93633) and 1.02 times larger than the probability of $P_{VAMF3\_L1}^{ZC\_E}$ (0.83865 against 0.81969);At t=11 h, the function of $P_{VAMF3\_\_L2}^{ZC\_E}\left(t\right)$ has a maximum equal to 0.185 and, at t=9 h, the function $P_{VAMF3\_\_L3}^{ZC\_E}\left(t\right)$ has a maximum equal to 0.008.

For $k_{E}=3\;$ (see Figure 20 and Figure 21):
The increase in the operating time t from 0 to 15 leads to an increase in the probabilities of $P_{VAMF3\_l2}^{ZC\_E}$ and $P_{VAMF3\_L3}^{ZC\_E}$ from 0 to 0.00945 and from 0 to 0.00333, respectively;At t=15 h, the probability of $P_{VAMF3\_L2}^{ZC\_E}$ is 2.84 times larger than the probability of $P_{VAMF3\_L3}^{ZC\_E}$ (0.00466 against 0.00124);At t=9 h, the probability of $P_{VAMF3\_L2}^{ZC\_E}$ is 3.77 times larger than the probability of $P_{VAMF3\_L3}^{ZC\_E}$ (0.00945 against 0.00333).

## 4. Case Study

### 4.1. Drone Fleet and IoD-Based Industrial Facility Monitoring System

#### 4.1.1. Structure of IoD SM

Let us consider the structure of a drone fleet and an IoD-based industrial facility monitoring system utilized in emergency mode (Figure 22). This structure is a special case of the type-T2 SM structure. The difference is that the IoD SM utilizes one UC only. The most vulnerable part of the IoD SM is the ComZC_FCC (a component of the FCC channel) [71,73,74,76,77] comprising drones for transmitting monitoring information from the ZC to the FCC. Thus, methods aimed at increasing the reliability of the ComZC_FCC require consideration.

#### 4.1.2. Reliability Model of FCC Channel

To improve the reliability of the ComZC_FCC, a structure of type ‘k-out-of-n’ [73], was proposed (see the RBD in Figure 23).

A ComZC_FCC with such a structure consists of *n* = 6 identical drones (DrnZC_FCC1, DrnZC_FCC2, …, DrnZC_FCC6) including four (*k* = 4) main drones (DrnZC_FCC1, DrnZC_FCC2, …, DrnZC_FCC4) and four (*n* − *k* = 6 − 4 = 2) standby (redundant) drones (DrnZC_FCC5 and DrnZC_FCC6). The ComZC_FCC remains in an operable state until three (*n* − *k* + 1 = 6 − 4 + 1 = 3) drones have failed. The ComZC_FCC can be considered as a series system with two (*n* − *k* = 6 − 4 = 2) redundant drones, each of which can replace any one of the failed operating drones. 

Assume that drones have an exponential time to failure. In this case, the reliability function for the ComZC_FCC can be written, considering [78], as
(30)$$P_{ComZC\_FCC}^{E}=\sum_{i=0}^{n-k}\frac{{\left(k2k_{E}\lambda_{dr}t\right)}^{i}}{i!}e^{-k2k_{E}\lambda_{dr}t}$$
where $\lambda_{dr}$ is the failure rate of the drone.

Thus, the reliability function of the FCC channel can be calculated as:(31)$$P_{FCC}^{E}=P_{ComZC\_FCC}^{E}P_{PrcZC\_FCC}=\sum_{i=0}^{n-k}\frac{{\left(k2k_{E}\lambda_{dr}t\right)}^{i}}{i!}e^{-k2k_{E}\lambda_{dr}t}\;e^{-\lambda k_{PrcZC\_FCC}t}.$$

Using Equation (31), some dependencies were obtained (Figure 24 and Figure 25)

The analysis of the dependencies obtained allowed us to draw the following conclusions.

For $\lambda_{dr}=0.001\frac{1}{h}$ (Figure 24):
At t=9 h, t=12 h, and t=15, the increase in the emergency coefficient $k_{E}$ from 1 to 9 leads to a decrease in the value of the reliability function $P_{FCC}^{E}$ from 0.99904 to 0.97100, from 0.99904 to 0.94181, and from 0.99824 to 0.90306, respectively;At $k_{E}=9,$ the value of the reliability function $P_{FCC}^{E}$ at t=9 h is 1.03 times larger than $P_{FCC}^{E}$ at t=12 h (0.97100 against 0.94181) and 1.08 times larger than $P_{FCC}^{E}$ at t=15 h (0.97100 against 0.90306).

For t=12 h (Figure 25):
At t=9 h, t=12 h, and t=15, the increase in the emergency coefficient $k_{E}$ from 1 to 9 leads to a decrease in the value of the reliability function $P_{FCC}^{E}$ from 0.99904 to 0.97100, from 0.99904 to 0.94181, and from 0.99824 to 0.90306, respectively;At $k_{E}=9,$ the value of the reliability function $P_{FCC}^{E}$ at t=9 h is 1.03 times larger than $P_{FCC}^{E}$ at t=12 h (0.97100 against 0.94181) and 1.08 times larger than $P_{FCC}^{E}$ at t=15 h (0.97100 against 0.90306).

### 4.2. The Equipment Monitoring System

#### 4.2.1. Principles and Structure

The equipment monitoring system of one industrial enterprise in Ukraine demonstrates the effect of the use of Industry 4.0 in KOEEBOX devices [72,79]. The KOEEBOX device is located in the middle of the power line and performs analyses of the electricity consumption dynamics. After receiving the data from the electricity grid, the device transmits them to the appropriate edge node for processing and subsequent aggregation and viewing of the PCG or CR. Viewing is possible through a special application for the KOEEBOX device. Table 5 shows the possibility of using the KOEEBOX device as a means of monitoring the different components and levels of the SM as a whole.

The digital twin technology with the IoT establishes a connection between the equipment and the CR. Additionally, it can be connected with cloud applications (PCG level). The energy performance monitoring device is available in the proposed equipment monitoring system. This device must be located in the middle of the power line so that it passes power to the end device. Hence, it performs power difference analysis and obtains status and usage statistics. The device transmits the data, after receiving them from the facilities, to the corresponding web server for further viewing and aggregation. Viewing is possible through a special system application. Figure 26a shows a block diagram explaining the monitoring of the energy consumption using sensors (ECS) for equipment verification (EC1—ECn).

In the proposed equipment energy-efficiency monitoring system, all operations are performed in real time both on the client side and on the device. The cloud part of the equipment energy-efficiency monitoring system should consist of three web services: the storage service; authentication service; and maintenance of client applications. Each service must run in its own isolated space and they must communicate with each other over the Internet. For the development of all services, it is suggested that one should use the JavaScript/TypeScript programming language, as well as WebSocket and HyperText Transfer Protocol (HTTP). The PostgreSQL database management system was chosen as the storage environment. All services must be designed according to the ECMAScript6 standard and SOLID design principles. Additionally, it is necessary to have an authorization mechanism for JavaScript Object Notation Web Token (JWT) [80].

#### 4.2.2. Processes and Algorithms of Monitoring

The storage service receives data from clients using the WebSocket protocol in real time, and also processes requests for statistical data on changes in energy efficiency over a certain period. Figure 26b shows the scheme of integration of the storage service into the overall system of monitoring the energy efficiency of the equipment. In this scheme, the storage service will act as a “digital twin” of the connected device and also reproduce the operation of the device in digital format. This component will have three interfaces for integration:A WebSocket gateway for devices;A WebSocket gateway for clients;An Application Programming Interface (API) gateway for HTTP Representational State Transfer (REST) for clients.

The WebSocket gateway for devices will be used to constantly “talk” to the device and maintain it, and in the future, to control the device through this gateway. The WebSocket gateway for clients is required to instantly transmit data received from a device to the end client, as well as for future device management. The API gateway for HTTP REST for clients will act as an accessible and easy-to-use statistics-generation interface, primarily required for the client application service, and there is a prospect of its integration with other services through this channel.

There are three main algorithms for a system for equipment monitoring. The first algorithm corresponds to the behavior of the service when the client interacts with the provided API gateway HTTP REST. This algorithm is classic for client–server architecture with built-in authorization checks [81], so all private resources will contain middleware that, in case of a mismatch, will return the corresponding error when performing the authorization check action of incoming requests. The second algorithm corresponds to the behavior of the service when the client interacts with the provided WebSocket gateway. The first three steps of the algorithm describe starting the program and reading the configuration files. The service stores the ID of the connected and authorized socket and waits for new data from the client-monitoring object named “device” after it verifies that the connected socket was authorized successfully. This socket ID save action is necessary to group sockets from a single client for the further filtering of client lookups and data transfers.

The next algorithm handled by the storage service is the algorithm that matches the behavior of the service when the device communicates with the provided WebSocket gateway. The first steps remained the same as in the previous two algorithms since these interfaces work in the same service. The next step, “Checking object data”, corresponds to checking the state of the corresponding device, which will be received from the authentication service via the API gateway HTTP REST. If the check is successful, the service stores the connected socket ID and waits for new data. The service performs a data validation process when the device sends data and, if successful, stores the data in the PostgreSQL repository; then, it forwards the data to the appropriate client sockets. Data from the device will be ignored if the verification fails [82,83].

#### 4.2.3. Experiment Results

Two types of system behavior were tested as an experimental task:The display of energy characteristics in real time;The display of energy characteristics in the past.

For the first experiment, a device emulator was used, which sent energy data to the system once every second. The result of this experiment has been described in [72].

For the second experiment, a device emulator was used for an hour, which also sent data every second. Figure 27 shows the result of this experiment. In this figure, the chart has more points due to the larger range of data displayed. Each point corresponds to the two-minute mean value of each parameter.

This system is described by the RBD in Figure 4.

## 5. Discussion

The distinctive feature of the proposed methodology is that of ensuring the dependability of systems on its developed base, which is achieved by combining the redundancy and diversity of various components and subsystems; these include sensors, means of communication and data processing, digital duplication formation, and decision support centers. The use of diversity reduces the risk of common cause failures, which can be caused by the influence of the external environment and the accumulation of failures, as well as cyberattacks.

The proposed SM structures, in comparison with existing works [53,54,63,71,73,76,84], provide higher reliability, as assessed using the reliability function (see Section 3.3). Additionally, these structures increase the survivability of the SM as an MSS (see Section 3.4). The mentioned benefits are achieved due to utilization of some kinds of redundancy and reconfigurability.

In addition, different structures of monitoring systems are proposed, depending on options for forming the models of digital duplicates using decision-making centers (CP, ECR, PCG, and FCC). It helps to select such structures, taking into account requirements for reliability and dependability, as well as characteristics of industrial facilities.

The set of proposed SM reliability models (Section 3) provides possibilities for the high-level quantitative calculation of indicators, the comparison of structure options, and a choice regarding their use. Moreover, a simulation was performed for selected values of the component reliability parameters. This makes it possible to determine the appropriate areas of the application, taking into account requirements and constraints, as well as the growth of failure rates due to the action of accident factors. One of the features of the proposed models is that they take into account the deterioration of monitoring system channels (at the ZC level) due to failures, and are presented as multi-state systems. The reliability of IoT devices that provide data for the DTs is taken into account in the SM reliability models. Their failure rate and battery life generally depend on the operation modes (including the modes of their possible periodic transfer to sleep mode). These modes are described by the corresponding coefficients when calculating the failure rates. Therefore, the selection of IoT devices is carried out while taking into account the requirements for SM reliability according to the proposed analytical dependencies.

The two examples for the application of monitoring systems complement each other according to the hierarchy levels of their representation in industrial facilities. For the enterprise equipment monitoring level (the first level of the industrial facility hierarchy), the first example of an equipment monitoring system based on the analysis of energy consumption dynamics is given. It can complement the traditional control and management methods in a fairly simple way, as well as form an additional component of the digital twin. This paper represents a significant extension of [72] with deeper details regarding the hierarchical building of monitoring systems.

The second example relates to more complex monitoring systems, which are part of Post-Accident Monitoring System and Automated Radiation Monitoring System. They are implemented using the UAV/FCC mobile subsystem and the PCG private cloud environment. The FCC subsystem redundancy option illustrates how its reliability can be increased and the composition of the UAV fleet can be selected.

## 6. Conclusions

The solving of research tasks enables us to make decisions about the construction and modernization of monitoring systems for complex industrial facilities, which increase their dependability and safety.

The results obtained are the basis for further developing the theory and practical applications of monitoring systems for industrial facilities within the framework of the implementation and improvement of Industry 4.0 principles.

In our opinion, the following areas of research and development are important and promising:The development of ontological models describing the intelligent UAV- and IoT-based monitoring systems in various situations and environmental conditions;The development of different separate and joint digital twin models for centers of decision making and for the implementation for optimal procedures of critical object recovery;The development and research of SM dependability models considering the extended taxonomy of hardware and software faults, recovery procedures and the location of the SM, as well as automated battery maintenance systems [84];The consideration of the cybersecurity aspects of the Internet of Drones for subsystem FCC and subsystem PCG for the assessment and assurance of SM reliability [85].

## Figures and Tables

**Figure 1 sensors-22-06444-f001:**
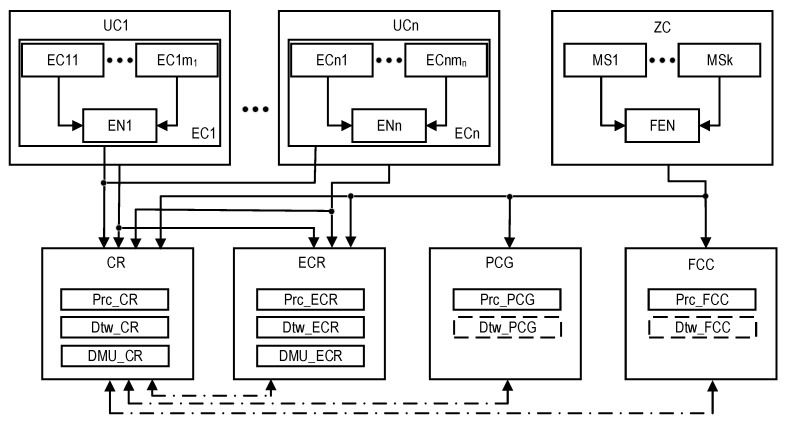
The structures of the industrial facility monitoring system: the T1-type structure which, in addition to the DTs for CR and ECR, requires DTs (depicted as dashed rectangles) for the PCG and FCC, and the T2-type structure which does not require them. The dot-dash arrows are utilized to show the channels for information exchange between the CR and other subsystems.

**Figure 19 sensors-22-06444-f019:**
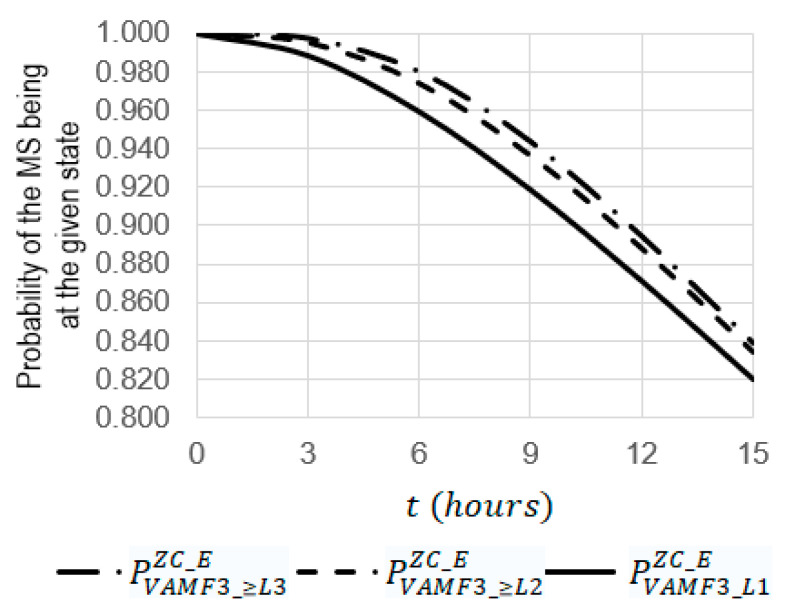
Dependencies of probabilities of $P_{VAMF3\_\geq L3}^{ZC\_E}$, $P_{VAMF3\_\geq L2}^{ZC\_E}$, and $P_{VAMF3\_L1}^{ZC\_E}$ on the operating time t at the emergency coefficient $k_{E}$ = 12.

**Figure 20 sensors-22-06444-f020:**
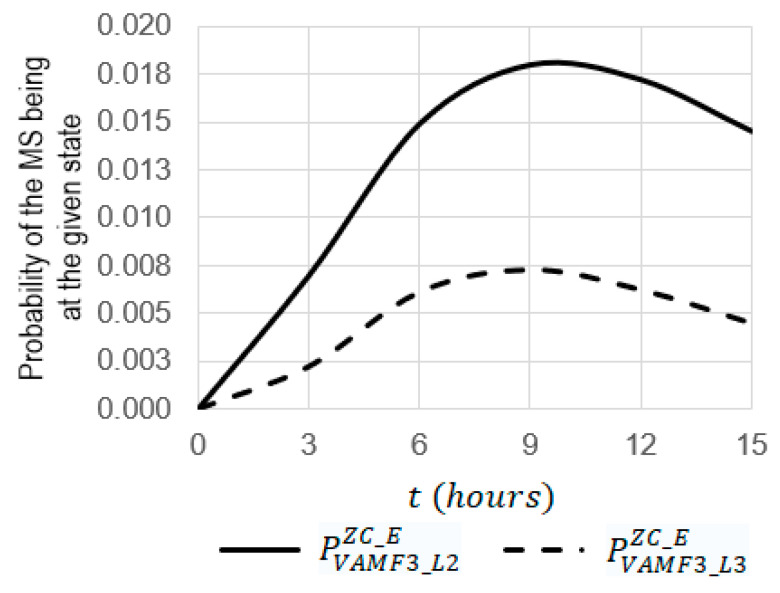
Dependencies of probabilities of $P_{VAMF3\_L2}^{ZC\_E}$ and $P_{VAMF3\_L3}^{ZC\_E}$ on the operating time t at $k_{E}$ = 12.

**Figure 21 sensors-22-06444-f021:**
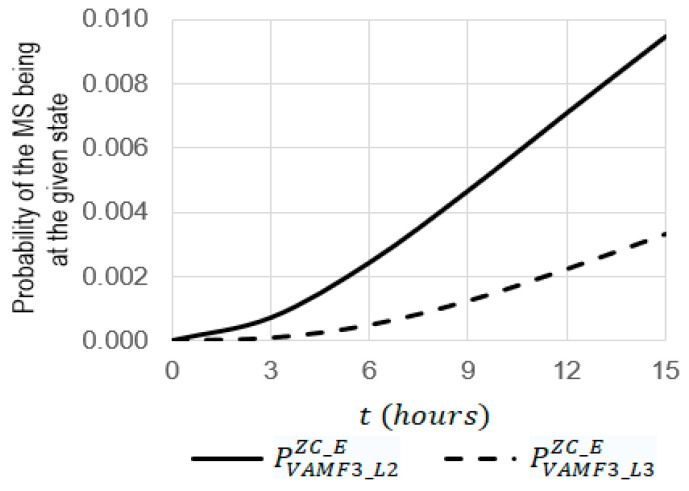
Dependencies of probabilities of $P_{VAMF3\_L2}^{ZC\_E}$ and $P_{VAMF3\_L3}^{ZC\_E}$ on the operating time t at the emergency coefficient $k_{E}$ = 3.

**Figure 22 sensors-22-06444-f022:**
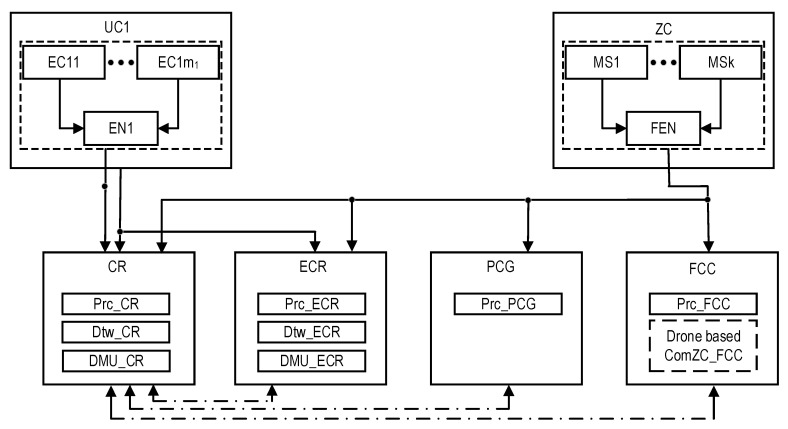
Structure of the IoD SM.

**Figure 23 sensors-22-06444-f023:**
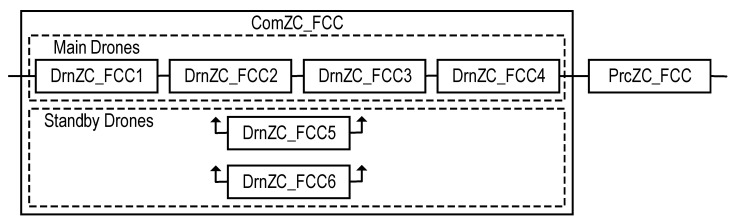
RBD of the FCC channel comprising the Prc_FCC and drone-based ComZC_FCC.

**Figure 24 sensors-22-06444-f024:**
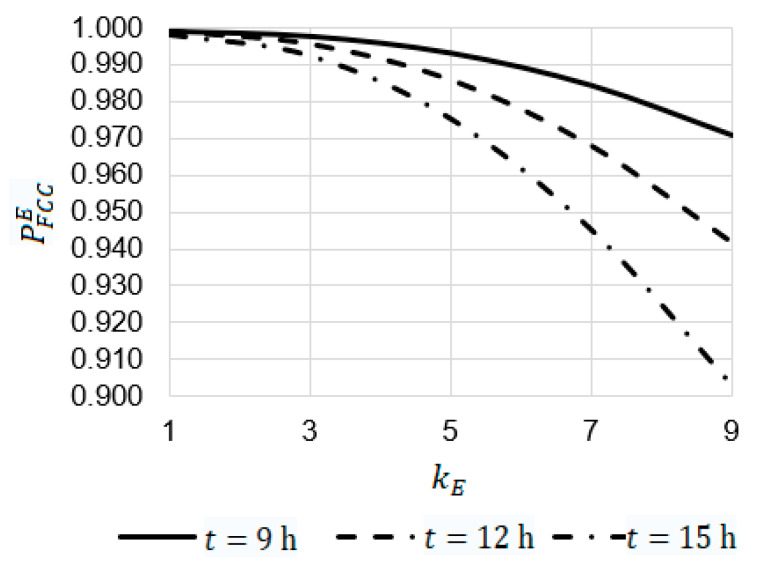
Dependency of the probability of $P_{FCC}^{E}$ on the emergency coefficient $k_{E}$ at a drone failure rate of $\lambda_{dr}=0.001\;h$.

**Figure 25 sensors-22-06444-f025:**
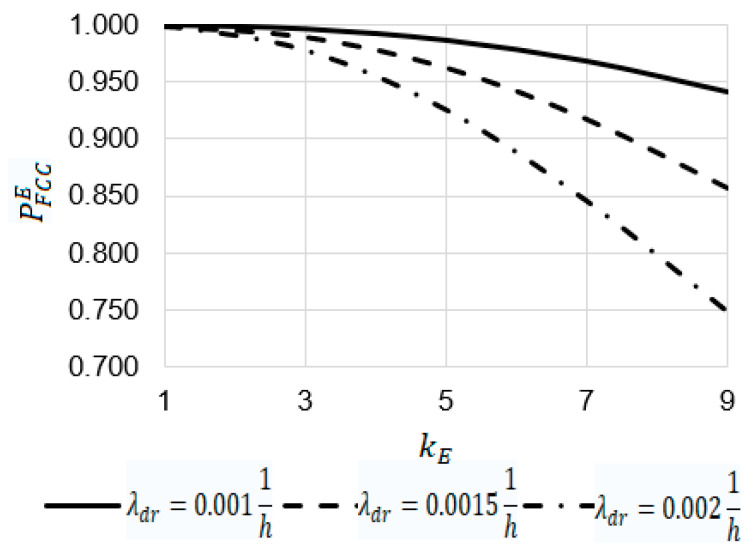
Dependence of the probability of $P_{FCC}^{E}$ on the emergency coefficient $k_{E}$ at the operating time t=12 h.

**Figure 26 sensors-22-06444-f026:**
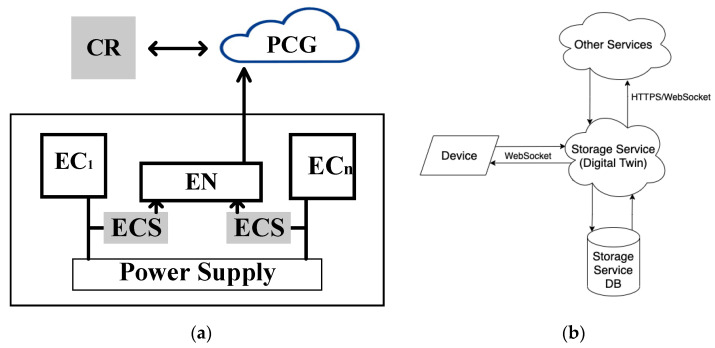
The flowchart with an explanation of the problem statement (**a**) and storage service integration scheme (**b**).

**Figure 27 sensors-22-06444-f027:**
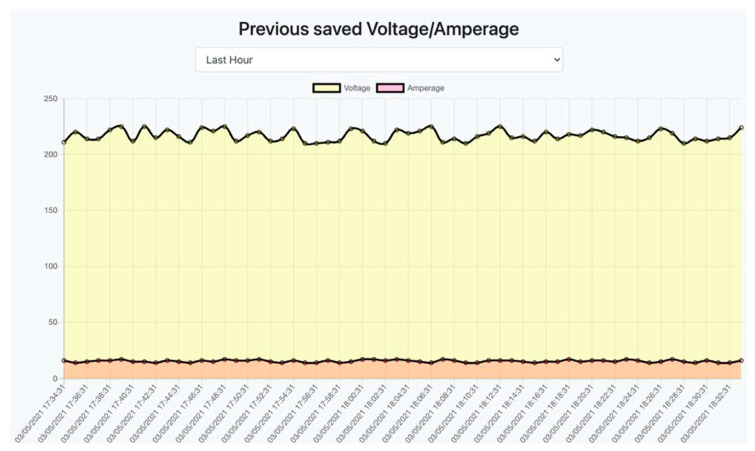
Displaying data for the last hour.

**Table 1 sensors-22-06444-t001:** Functions of centers.

Mode	Centers of Control and Monitoring			
	CR	ECR	FCC	PCG
Normal	Monitoring EC, UC, ZC	Monitoring UC, ZC	Waiting (self-maintenance)	Waiting (self-maintenance)
Emergency	Monitoring EC, UC, ZC	Monitoring UC, ZC	Monitoring ZC	Monitoring ZC

**Table 2 sensors-22-06444-t002:** The capacity of monitoring functions.

Mode	Level	Centers of Control and Monitoring			
		CR	ECR	FCC	PCG
Normal	EC	Full	None	None	None
	UC	Full	Full	None	None
	ZC	Full	Full	None	None
Emergency	EC	Full	None	None	None
	UC	Full	Full	None	None
	ZC	Full	Full	Full/Partial	Full/Partial

**Table 3 sensors-22-06444-t003:** Specification of SM structures and configurations in normal and emergency modes.

Mode	Source of Information	Centers and Subsystems	Type of Structure
Normal	EC	CR	T1
Normal	EC	CR	T2
Normal	UC	CR, ECR	T1
Normal	UC	CR, ECR	T2
Normal	ZC	CR, ECR	T1
Normal	ZC	CR, ECR	T2
Emergency	EC	CR	T1
Emergency	EC	CR	T2
Emergency	UC	CR, ECR	T1
Emergency	UC	CR, ECR	T2
Emergency	ZC	CR, ECR, PCG*, FCC*	T1
Emergency	ZC	CR, ECR, PCG*, FCC*	T2
Emergency	ZC	CR, ECR, PCG, FCC*	T1
Emergency	ZC	CR, ECR, PCG, FCC*	T2
Emergency	ZC	CR, ECR, FCC, PCG*	T1
Emergency	ZC	CR, ECR, FCC, PCG*	T2
Emergency	ZC	CR, ECR, PCG, FCC	T1
Emergency	ZC	CR, ECR, PCG, FCC	T2

**Table 5 sensors-22-06444-t005:** Use of the KOEEBOX device as a means of monitoring the energy consumption of SM-EC.

Name	Possibility of Using the Device KOEEBOX	How It Can Be Used	Working Principle
CR/ECR	+	Control and monitoring of the equipment of the center	Collects and analyzes statistics on power supply of equipment
PCG	+	Server monitoring	Monitors server power status
FCC	+	Control and monitoring station	Monitors control panels status

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
