# Peer review of "UAV and IoT-Based Systems for the Monitoring of Industrial Facilities Using Digital Twins: Methodology, Reliability Models, and Application"

_sensors, 2022, doi:10.3390/s22176444_

Round 1
Reviewer 1 Report
The authors proposed suggests a scheme for building two-mode supervising techniques (SMs) of industrial facilities and adjacent territories combining unmanned aerial vehicles, Internet of Things, and digital twins (DTs) technologies. The idea is very interesting. However, I have the following concerns.
1. A check on English correction is required.
2. Recent UAV-assisted IoT-based applications are missing. Some are mentioned as follows.
-> "Joint Optimization of Trajectory and Resource Allocation in Secure UAV Relaying Communications for Internet of Things," in IEEE Internet of Things Journal, doi: 10.1109/JIOT.2022.3151105.
-> "FBI: A Federated Learning-Based Blockchain-Embedded Data Accumulation Scheme Using Drones for Internet of Things," in IEEE Wireless Communications Letters, vol. 11, no. 5, pp. 972-976, May 2022, doi: 10.1109/LWC.2022.3151873.
3. Add a summary table for symbols.
4. A summary table for state-of-the-art containing their limitations.
5. The motivation behind applying DT and UAV is not clear. Overall contribution is also not clear.
6. What is IoD? What's the difference between Drone and IoD-based industrial facility monitoring systems?
7. Too many shortcodes are used which makes the paper difficult to read. Please simplify the paper.
8. Add a system model to describe the paper in simple words.
9. A comparison with existing works is missing.
10. Did the authors consider the limited battery and computation of UAV? How will UAV need to be served in the proposed scheme?
11. To get connected with DT, a continuous flow of data is needed to be maintained. IoT is both generating data and transmitting it continuously. Won't it cost the battery of IoT devices which is limited?
Author Response
Dear Reviewer,
Please see the attachment.
Kindest regards,
Authors

Reviewer 2 Report
The authors propose a method for monitoring of facilities using digital twins (DTs), unmanned aerial vehicles (UAVs), and IoT defining also the reliability of the developed methodology
The paper is sufficiently well written even if there are some sentences that need further explanation.
The use of acronyms is extensive, hence I suggest limiting the abbreviations only to the cases in which are used more than once and adding a list to help the reader.
The overall structure of the paper is solid, but there are parts that can be shortened/expanded.
A list of the main comments/doubts/request is as follows:
1. Introduction
This is fairly solid and the paper present a wide range of works on the subject proposed. Nevertheless, I believe that some references to important aspects of DT is missing, in particular:
a) When speaking of process monitoring in Industry 4.0, it is also important to underline how recent applications can be achieved with appropriate strategies of data collection and cloud storage systems (see “A cloud-based monitoring system for performance assessment of industrial plants (2020)”)
b) Another use of DT for monitoring is in those situations in which real-time measurements are not feasible due to economical or technological factors. In these cases, the DT can be used to predict such measurement at a change in operating conditions of the operations/facility and the operator is able to check whether all the predictions are in the safe zone (see “Development and Assessment of an Intelligent Compaction System for Compaction Quality Monitoring, Assurance, and Management (2022)”, “A rigorous simulation model of geothermal power plants for emission control (2020)”)
c) Finally, monitoring can also be performed via DT by checking if the normal operation variables are maintained in the correct range when performing optimization techniques that calculate the optimal economic management of facilities (see “Bridging the Gap in Technology Transfer for Advanced Process Control with Industrial Applications (2022)”, ”Optimally Managing Chemical Plant Operations: An Example Oriented by Industry 4.0 Paradigms." (2021) and/or "Implementation of an Industry 4.0 system to optimally manage chemical plant operation." (2020))
- the acronym NPP is not defined.
2. Methodology of building SMs
- Figures 1 and 2 are almost identical. The authors should maintain only figure 1 and describe it better in the caption in which case is T2. In addition, the caption should also evidence the different types of lines (dots, dashed, dash-dot,..) that are in the figure so that the reader can fully appreciate it and link them to the text.
- Table 3: define modes Normal/Nominal and Emergency, there are already many acronyms and at least these can be extensively written
3. System of Monitoring Reliability Models
- How do you assure the compatibility of different DTs in each SM subsystem? Or did I misunderstand the first sentence of 3.1?
-bottom of page 9: remove the dotted list and rephrase it as a paragraph
- at 3.3.1: the notation explanation should be reduced to make the reading lighter; e.g. “P_j is the reliability function of j”, “k_l are the coefficients…..” and then refer explicitly to Table 4.
- 3.3.2 can be moved in Appendix since there is no comment and only equations
- Figures 15 and 16 are identical in label, caption, and description but they are different in the dashed line, how is this possible? What does it change that it is not mentioned explicitly in the captions?
and so for Figures 17 and 18… 19 and 20 … 21 and 22.. All these figures should be reduced to subplots by specifying what changed between them.
- The comments on the figures should be general and systematized not copied and pasted many times by changing the numbers. This is what happens in the last 4 paragraphs of 3.3
Author Response

(The authors gave the same response as above.)

Round 2
Reviewer 1 Report
I am recommending to accept this paper.
Author Response
Dear Reviewer,
thank you for your feedback!
Kindest,
Authors
Reviewer 2 Report
The authors have critically addressed all of my suggestions and doubts thus improving the quality of the manuscript.
I suggest a final revision of the text to check for minor typos.
Author Response
Dear Reviewer,
Thank you for your feedback. We corrected the typos.
Kindest regards,
Authors.